# An In Vivo Screening Model for Investigation of Pathophysiology of Human Implantation Failure

**DOI:** 10.3390/biom13010079

**Published:** 2022-12-30

**Authors:** Hitomi Nakamura, Tadashi Kimura

**Affiliations:** Department of Obstetrics and Gynecology, Osaka University Graduate School of Medicine, Suita 565-0871, Japan

**Keywords:** implantation, implantation failure, uterine receptivity, female infertility, HVJ-E vector, gene delivery system, NF-kappa B, Stat-3

## Abstract

To improve current infertility treatments, it is important to understand the pathophysiology of implantation failure. However, many molecules are involved in the normal biological process of implantation and the roles of each molecule and the molecular mechanism are not fully understood. This review highlights the hemagglutinating virus of Japan (HVJ; Sendai virus) envelope (HVJ-E) vector, which uses inactivated viral particles as a local and transient gene transfer system to the murine uterus during the implantation period in order to investigate the molecular mechanism of implantation. In vivo screening in mice using the HVJ-E vector system suggests that signal transducer and activator of transcription-3 (Stat-3) could be a diagnostic and therapeutic target for women with a history of implantation failure. The HVJ-E vector system hardly induces complete defects in genes; however, it not only suppresses but also transiently overexpresses some genes in the murine uterus. These features may be useful in investigating the pathophysiology of implantation failure in women.

## 1. Introduction

Implantation failure has long been considered as a major problem in assisted reproductive technology (ART) treatment. This is especially so since the technology for preimplantation genetic testing for aneuploidy (PGT-A) has been implemented worldwide for ART treatment [1,2,3]. Uterine receptivity has been clinically evaluated using serum progesterone (P4) levels, ultrasonographic endometrial thickness, and histological dating by endometrial biopsy. A discrepancy between the hormonal milieu and endometrial morphology and/or function has been suggested as an important cause of implantation failure [4,5,6,7], despite the lack of a consensus diagnostic criteria to date.

Implantation can only occur within a very short period, known as the “implantation window”, which is approximately 24 h in the rodent uterus [8]. Even though the implantation period is brief, the uterus undergoes multiple complex orchestrated changes [9,10]. Implantation is primarily organized by estrogen (E2) and P4. E2 stimulates the proliferation of uterine endometrium. Following ovulation, the corpus luteum secretes P4, and it changes the uterine endometrium in the secretory phase. The uterine endometrium ceases epithelial proliferation and begins to undergo differentiation during the secretory phase. P4 priming, which is superimposed with E2 priming, leads to the uterine endometrium entering the receptive phase. The pre-receptive (preparatory) phase occurs on days 1–3 in mice and in the early luteal phase in women (approximately one week directly after ovulation). Blastocysts implant the uterine endometrium only during the receptive phase (day 4 in mice, mid-luteal phase in women) but not in the pre-receptive phase [9,11]. After the receptive phase, the uterine endometrium shifts to the non-receptive phase. E2 and P4 are essential for the initiation of the receptive phase of the uterine endometrium [9,12]. For infertility treatment, both E2 and P4 have been and assessed and used to support and optimize implantation in ART treatment for a long time. Moreover, the dosage form and route of administration of E2 and P4 have been optimized however, the pregnancy rate is still not satisfactory. A systematic review showed no difference in the clinical pregnancy rate per woman after frozen-thawed embryo transfer between the natural menstrual cycle and various controlled regimes using exogenous E2 and P4 supplements [13]. E2 and P4 have the essential primary roles in preparing the uterine endometrium for the receptive phase. However, they do not have total control over the state of uterine receptivity.

During and before the receptive phase in the uterine endometrium, many molecules, including adhesion molecules, growth factors, and cytokines, are involved in the biological process of implantation in this short period and are highly organized [9,14]. However, their roles in uterine receptivity and implantation and their underlying molecular mechanisms are not fully understood. Many transgenic mouse models have been used to assess their roles in uterine receptivity and implantation. However, this did not always help because their constitutive deletion led to embryonic lethality or other systemic deficiencies. Unlike other organs, uterine endometrium changes their structure and roles within a short period. Implantation is a very short period; however, uterine endometrium is controlled by the hierarchical directives. To investigate the hierarchical directives that orchestrate the uterine endometrium, a transient and local in vivo gene transfer system, rather than long-term and systematic gene manipulation, to the murine uterus would be helpful. In this review, we introduce an in vivo screening mouse model that uses a transient and local gene transfer system to understand the pathophysiology of implantation failure in women.

## 2. HVJ-E Vector System as a Gene Delivery System

Endocytosis-mediated delivery systems, such as liposomes, have been widely used as gene delivery systems. Their weakness, however, is the degradation of nucleotides and other molecules before they reach the cytoplasm [15]. Fusion-mediated delivery systems have been developed to avoid this problem. As fusion-mediated delivery systems, the development of paramyxovirus vectors for gene delivery have received considerable attention, as these viruses show robust fusion activity with the cell membrane and replicate in the cytoplasm without integration into the DNA [16].

The mouse parainfluenza virus, hemagglutinating virus of Japan (HVJ), also known as Sendai virus, is a paramyxovirus [17]. HVJ binds to cell surface sialic acid via its hemagglutinin neuraminidase (HN) protein and promotes membrane fusion via its fusion (F) protein [18]. The HVJ envelope (HVJ-E) vector employs a native HVJ shell with powerful cell-fusion properties. The viral RNA is completely degraded and inactivated by UV irradiation and subsequent treatment with mild detergent to incorporate plasmid DNA, proteins, synthetic oligonucleotides, and drugs inside the HVJ-E vector [19,20]. The HVJ-E vector system can deliver genes to a wide range of cells, both in vitro and in vivo [19,20,21,22,23,24].

## 3. A Transient and Local In Vivo Gene Transfer System to Murine Uterine Endometrium Using HVJ-E Vector

To investigate the hierarchical directives that orchestrate the uterine endometrium in mice, we optimized the HVJ-E vector system as a transient and local in vivo gene transfer system for the murine uterus [22]. Mice on day 1.5 post-coitum (p.c.; the morning on which vaginal plugging was first observed was designated as day 0.5 p.c.) were anesthetized and subjected to a laparotomy to expose the uterus. The HVJ-E vector containing plasmid DNA or oligodeoxynucleotides (ODNs), total 25 μl in human tubal fluid medium, was injected slowly into the uterine cavity using a 30-gauge needle from the oviduct side of the utero-tubal junction, and the injection sites and cervix were clamped for 10 min using vascular clips for small animals (Figure 1).

Gene transfer using the HVJ-E vector is approximately 120-fold or 17-fold more efficient than the introduction of the same dose of DNA using liposomes or HVJ-liposomes, respectively [20,22]. Gene transfection of the uterus using the HVJ-E vector is transient. The maximum efficiency of gene transfection was achieved 24 h after gene transfer. On day 3 following gene transfer, the transferred gene expression was approximately 50% lower than that of 24 h after transfection [22]. On day 5 after gene transfection of luciferase-pcDNA3 plasmid DNA (pcDNA3-LUC-GL3), luciferase activity was not detected (Figure 2) [22].

When plasmid DNA (pcDNA3-LacZ) was transferred to the uterus, the transferred gene was distributed in both the luminal and granular epithelium. In contrast, when FITC-labelled ODNs (20 mer in a random sequence) were transferred using the HVJ-E vector, it was distributed not only in the endometrial epithelium but also in the stromal cell layer and a few myometrial cells [22].

HVJ-derived protein (F-protein) disappeared from the uterine cavity within 72 h of transfection [22]. The HVJ-E vector had no effect on pregnancy rate, course of pregnancy, litter size, fetal growth in utero, or parturition. Furthermore, it did not transfect the exogenous gene to the fetus [22]. The HVJ-E vector system can also be applied to the murine uterus during the mid-late term of pregnancy without disturbing the pregnancy [25].

For in vivo uterine gene transfection in animal models, various methods have been reported, including liposomes [26,27,28,29], adenovirus [30], retrovirus [31], electroporation [32], bionanocapsule (BNC) comprising a hepatitis B virus envelope L-protein particle [33], and HVJ-E vector [20,22,24,25,34,35]. However, only a few studies have assessed the efficiency of gene transfection and distributions of transferred genes. To our knowledge, HVJ-E vector system is one of the most efficient gene delivery systems to the murine uterus.

## 4. An In Vivo Screening Model

Many molecules have been reported to be expressed in the uterine endometrium during implantation [9]. These molecules are under strict temporal and spatial regulation within a short period, although the biological process of implantation orchestrates complex changes. To maintain this strict temporal and spatial regulation, we hypothesized that a few multipotent transcription factors may regulate the transcription of various molecules that initiate the cascade of biochemical modification of the endometrium to open the implantation window. We used the HVJ-E vector system in a mouse in vivo screening model to modulate the function of endometrial signaling molecules during the implantation window without disturbing the course of pregnancy.

We focused on nuclear factor-kappa B (NF-κΒ) and signal transducer and activator of transcription-3 (Stat-3) as multipotential transcription factors that might regulate the transcription of various molecules.

To assess their function in uterus during implantation, local and transient suppression of these transcription factors in the uterus was induced using the HVJ-E vector system. For the transient suppression of NF-κB and Stat-3, a dominant negative mutant inhibitor κBα of NF-κB (pcDNA3-IκBαM, Clontech #631923) and Stat-3 (Y705F, pcDNA3-Stat-3-Y705F, RIKEN DNA Bank #RDB02354) or double-stranded decoy ODNs (Table 1) were transferred into the uterine cavity using HVJ-E vector on day 1.5 p.c. On day 15.5 p.c., the viable fetuses in utero in each group were assessed (Figure 3). All mice became pregnant in the groups transferred with control plasmid DNA (pcDNA3-LUC-GL3), control double-stranded scramble decoy ODNs, and Stat-3 decoy (#1 in Table 1, single cis-binding sequence). However, in the pcDNA3-IκBαM and NF-κB decoy-transferred groups, only 70–80% of mice became pregnant, and in the pcDNA3-Stat3-Y705F and Stat-3 decoy (#2 in Table 1, double cis-binding sequences)-transferred group, approximately 70–75% of mice lacked viable fetuses.

## 5. NF-κB Activation Determines the Timing of Implantation

NF-κB is a member of a protein family whose members, NF-κB1 p50, NF-κB2 p52, Rel A (p65), Rel B, and c-Rel, form heterodimers. In the resting form, NF-κB is inactivated in the cytoplasm by its endogenous inhibitor IκB [36]. The signal to stimulate NF-κB first results in the phosphorylation of IκB, leading to the release of NF-κB from the inactive NF-κB-IκB complex. NF-κB is a transcription factor involved in many inflammatory and immune responses. NF-κB is activated in the pro-estrus and estrus phases in the uterine endometrium of nonpregnant mice [37]. Starting on day 1.5 p.c., NF-κB is gradually activated in the pregnant uterus every day and the activation continues throughout the implantation period [37]. The immunoreactivity of p50 and p65 was predominant in the endometrial epithelium and weaker in the endometrial stromal cells. Knockout of p65 in mice results in embryonic lethality, whereas targeted disruption of p50 causes no developmental abnormalities [38,39].

Using the HVJ-E vector system, the transfection of pcDNA3-IκBαM or NF-κB decoy on day 1.5 p.c. suppressed approximately 50% of NF-κB activity compared to that in the control group on day 4.5 p.c.; NF-κB activity recovered to the control level on day 5.5 p.c. [35]. At day 4.5 p.c., the implantation site was not observed after an intravenous (i.v.) injection of 0.5% Evans Blue [40], but on day 6.0 p.c., the uterus formed an epiblast and a primitive endoderm (development stage 6, equivalent to day 4.0 p.c. by Theiler [41]). This transient and local suppression of NF-κB activity led to a significant delay in implantation and a delayed delivery date (Figure 2), but it did not affect litter size per mouse or birth weight. During implantation, p50 and p65 proteins were predominantly expressed in the luminal and glandular epithelium. The transfection of pcDNA3-IκBαM with the HVJ-E vector system suppressed NF-κB activity predominantly in the luminal and glandular epithelium. In contrast, the transfection of NF-κB decoy suppressed NF-κB activity in the luminal and glandular epithelium as well as the stromal cells layer. Both pcDNA3-IκBαM and NF-κB decoy-transfection groups showed similar rates of implantation failure. This transient and local suppression of NF-κB significantly decreased leukemia inhibitory factor (LIF) mRNA expression in the uterus on days 3.5 and 4.5 p.c., whilst the expression of cyclooxygenase-2 (Cox-2) and Hoxa-10 mRNA did not differ. The overexpression of LIF significantly increased the number of implantation sites on day 4.5 p.c. in the pcDNA3-IκBαM transferred mice (mean ± SD, 4.9 ± 0.9, range: 1−9) compared to the control overexpression group (pcDNA3-LacZ + pcDNA3-IκBαM, 0.7 ± 0.42, range: 0−3); however, it remained significantly lower than in the negative control mice (pcDNA3 + pcDNA3-LacZ transferred mice, 11.6 ± 0.4, range: 10–13). NF-κB activation determines the timing of implantation, at least in part, by controlling LIF expression (Figure 4) [35].

## 6. Interaction between Ovarian Hormones (E2 and P4) and NF-κB Activity

NF-κB plays different roles during the three stages of pregnancy which are (i) establishment; (ii) maintenance of pregnancy; and (iii) labor [44,45,46,47]. During the establishment of pregnancy, NF-κB plays important roles in opening the implantation and also regulates trophoblast invasion [47]. As previously described, pregnancy events are primarily coordinated by E2 and P4. NF-κB activation is also related to these hormones. E2 and estrogen receptor (ER) modulate NF-κB activity via both genomic and non-genomic processes, which are membrane-initiated steroid signaling [48] and actions [49]. E2 can rapidly activate NF-κB via non-genomic pathways [49]. However, many studies have reported that ER inhibits NF-κB activity in E2-dependent cells [50,51,52,53,54,55,56,57,58,59]. While the very rapid activation of NF-κB by E2 has been demonstrated in the uterus in vivo [60] and in other cells, including endothelial cells, cardiac myocytes, and splenocytes [49], the inhibition of NF-κB by E2 occurs much slower than the rapid response [49]. Moreover, ERα cooperates with NF-κB to activate other promoters without inhibiting the actions of NF-κB [50]. The molecular mechanism of ER and NF-κB activity is complicated. The mechanism may have negative feedback and other transcription pathways. The mechanism of NF-κB activity via E2 and ER is not fully understood. Numerous studies on various cells other than the uterine endometrium [47,61,62,63] have reported that activation of NF-κB by administration of P4 withdraws and suppresses NF-κB activity. Mutual repression between progesterone receptor (PR) and NF-κB in tissue targeted by P4 [47,63] has been reported. In women, E2 and P4 withdrawal occurs during menstruation and parturition. NF-κB is activated in the uterine endometrium during the perimenstrual phase and labor [47]. However, in mice, NF-κB is activated before labor [47] and at proestrus and estrus in the non-pregnant period [37]. Similar to women, E2 and P4 withdrawals occur during parturition in mice. Circulating levels of E2 and P4 during the estrous cycle in mice are slightly different from those in women during the perimenstrual phase. At proestrus stage, an increase in circulating levels of E2 and P4 is observed; however, the circulating levels of E2 decrease and withdraw at estrus [37], increase at diestrus, and peak at proestrus. However, the circulating levels of P4 increase at diestrus and peak at metestrus [37]. In mice, mating, which induces ovulation [64] occurs within a limited period of estrus. During early stage of pregnancy in mice, circulating levels of P4 increase on day 2–6 p.c. and are maintained at high levels throughout pregnancy [65]. Similar to women, during pregnancy, there is an increase in circulating levels of P4. In mice, PR mRNA levels increase on day 1.5 p.c. and the expression of PR protein reaches the maximum on day 3.5 p.c. [11,12]. Similar to women, PR expression in the uterine endometrium of mice diminishes on day 4.5 p.c. before embryo implantation [66,67,68]. However, NF-κB activity in the uterine endometrium starts to rise on day 1.5 p.c. until at least day 6.5 p.c. [37], which is the stage of trophoblast invasion. Subsequently, during the maintenance of pregnancy, decreased activation of NF-κB in women and mice until labor period is observed [47]. NF-κB regulates the maternal inflammatory response for opening implantation and trophoblast migration and invasion [35,47]. However, E2, P4, and their receptors’ expression do not fully explain how NF-κB activity is controlled in the uterine endometrium during implantation.

## 7. Stat-3 Regulates Blastocyst Attachment and Decidualization

STATs are a family of latent cytoplasmic proteins that transmit extracellular signals to the nucleus [69]. Stat-3 is activated by glycoprotein 130 (gp130), a subunit of the receptor for the interleukin-6 (IL-6) family of cytokines, including LIF and IL-11 [70]. Among the cytokine family members, only LIF and IL-11 receptor-α knockout in mice have an implantation failure phenotype [42,71,72]. LIF expression peaks shortly before the implantation window, whilst IL-11 expression peaks on day 5.5–7.5 p.c. [71]. Stat-3 is activated immediately prior to implantation in the uterine endometrium of mice and rats. Stat-3 deficiency in embryos is lethal [43].

In our study, the transfer of pcDNA3-Stat-3-Y705F or Stat-3 decoy #2 with double cis-binding sequences on day 1.5 p.c. suppressed Stat-3 activity in the uterus by approximately 50% on day 5.0 p.c. (Figure 2). It inhibited blastocyst attachment and impaired decidualization, which is an indispensable process for implantation and results in <30% implantation with normal progesterone levels [34].

Is Stat-3 activity vital to uterine receptivity? We aimed to investigate whether overexpression of Stat-3 in the uterine endometrium allows blastocysts to implant during pro-estrus, which is certainly not an implantation period. A constitutively active form of Stat-3 (Stat-3-C, Addgene plasmid #8722, pcDNA3-Stat-3-C), wild-type Stat-3 (pcDNA3-Stat-3, Addgene plasmid #8706), or control plasmid DNA (pcDNA3-LUC-GL3) was transferred into the uterine cavity during pro-estrus using the HVJ-E vector system. Seven to twelve blastocysts per horn were transferred into the uterine cavity 48 h after gene transfection. On day 2 after blastocyst transfer, the number of implantation sites per horn in the Stat-3-C group (n = 27, mean ± SD, 3.6 ± 0.4, Shapiro–Wilk normality test and Wilcoxon’s rank-sum test, *p* < 0.001) was significantly higher than those in the LUC-GL3 (n = 15, 0.5 ± 0.2) and wild-type Stat-3 (n = 10, 0.600 ± 0.4) groups.

To investigate the role of *Stat-3* in the uterine endometrium during implantation, studies that used conditional knockout mice with *Stat-3* gene only in *PR*-positive (*PR^cre/+^ Stat-3^f/f^*; *Stat-3^d/d^*) [73,74] and *Wnt7a*-positive (*Wnt7a ^cre/+^ Stat-3^f/f^*; *SW^d/d^*) [75] cells have been published. Lee et al. reported a similar number of retrieved oocytes in *Stat-3^d/d^* and *Stat-3^f/f^* mice after superovulation treatment [73]. However, Sun et al. reported a relatively smaller number of blastocysts in the uteri of *Stat-3^d/d^* mice than in the uteri of *Stat-3^f/f^* mice on day 4 p.c. [74]. Additionally, all the mice showed implantation failure owing to the lack of blastocyst attachment and impaired decidualization. During implantation, a strong expression of phosphor-Stat-3 is observed in the luminal and glandular epithelium as well as in the decidual stromal cells at the implantation site [73]. In *SW^d/d^* mice, *Stat-3* gene is conditionally inactivated in the uterine epithelium; however, it is retained in the stromal cells [75]. However, *SW^d/d^* mice showed drastically reduced decidual responses with suppressed stromal cell proliferation and differentiation [75]. As previously described, plasmid DNA *(pcDNA3)* gene transfected with HVJ-E vector system was distributed mainly in the luminal and glandular epithelium [22]. On the other hand, *ODNs* gene transfected with the HVJ-E vector system was distributed not only in the endometrial epithelium but also in the stromal cells layer, and a few myometrial cells [22]. The distributions of the transfected pcDNA3-Stat-3-Y705F and *Stat-3* decoy #2 in the uterus were different. Although *Stat-3* decoy #2 worked in the whole uterus, the transfection of pcDNA3-Stat-3-Y705F can only suppress *Stat-3* activity in the luminal and glandular epithelium. However, these two groups showed comparable suppression rates of implantation. These results suggest that epithelial *Stat-3* controls stromal function via paracrine mechanisms. LIF, a downstream estrogen target, is essential for implantation in mice [9,42,70]. LIF signal is transduced via gp130, a transmembrane protein, to either STAT-1/3 or Src homology region 2 domain-containing phosphatase 2 (SHP2)/extracellular signal-regulated kinase (ERK) pathways. Constitutive inactivation of gp130 induced embryonic lethality in mice. Transgenic knock-in mice with defective gp130-mediated STAT-1/3 signaling showed implantation failure [76]. Conditional knockout mice with *gp130* gene only in the *PR*-positive (*PR^cre/+^ Gp130-3^f/f^*; *Gp130-3^d/d^*) cells were infertile because of implantation failure, although a similar number of blastocysts was observed in mice with both *Gp130-3^f/f^* and *Gp130-3^d/d^* genes [74].

## 8. Interaction between Ovarian Hormones (E2 and P4) and Stat-3 Activity

Stat-3 is not a direct target of P4 regulation. However, it is an important collaborator of P4 signaling during the early stage of pregnancy [77]. The transient and local suppression of *Stat-3* activity in the uterus during implantation showed no significant change in serum levels of P4 and PR mRNA expression in the uterus on day 5.0 p.c. [34]. Sun et al. reported that serum levels and expression pattern of E2 and P4; and the intensity of ER and PR in *Stat-3 ^d/d^* mice were similar to those in *Stat-3^f/f^* mice on day 4 [74]. In contrast, Lee et al. reported a significant suppression of PR-expressing stromal cells on day 3.5 and 4.5 p.c in *Stat-3^f/f^* mice [73].

Homeobox A10 *(Hoxa10)* is a P4-responsive gene. The expression level of *Hoxa10* mRNA in the uterus on day 4 of pregnancy was lower in *Gp130-3^d/d^* than in *Gp130-3^f/f^* mice [74]. In *Stat-3^d/d^* mice, the expression level of *Hoxa10* mRNA in the uterus was almost undetectable [74]. Additionally, mRNA expression levels in other PR target genes, such as Indian hedgehog *(Ihh)* and amphiregulin *(Areg)*, in the uterus on day 4 of pregnancy in *Gp130-3^d/d^* and *Stat-3^d/d^* mice were lower than those in *Gp130-3^f/f^* and *Stat-3^f/f^* mice [74]. The mRNA expression of PR target genes, such as cytochrome P450, family 26, subfamily A, polypeptide 1 (*Cyp26a1*), follistatin (*Fst*), *Areg*, lipoprotein receptor-related protein 2 (*Lrp2*), and IL-13 receptor, α2 (*Il13ra2*), was significantly downregulated after administration of P4 (1 mg) in ovariectomized *Stat-3 ^d/d^* mice than in *Stat-3^f/f^* mice [73]. However, in one of the P4-responsive genes, histidine decarboxylase (*Hdc*), mRNA expression in the uterus on day 4 of pregnancy was similar in *Gp130-3^d/d^*, *Stat-3^d/d^*, *Gp130-3^f/f^*, and *Stat-3^f/f^* mice [74]. In contrast, lactoferrin (*Ltf*) and mucin 1 (*MUC1*), which are E2-responsive genes, were upregulated in *Gp130-3^d/d^* and *Stat-3^d/d^* mice [74]. High expression of *Ltf* mRNA was observed in the luminal and glandular epithelium of *Gp130-3^d/d^* and *Stat-3^d/d^* mice. However, *Ltf* mRNA was not detected in *Gp130-3^f/f^* and *Stat-3^f/f^* mice [74]. It is considered that MUC1 works as a barrier to blastocyst attachment to the luminal epithelium because it is significantly expressed in the uterine epithelium on day 1 of pregnancy, and it decreases on day 4 of pregnancy. *MUC1* immunostaining showed a strong expression in the apical membrane of the luminal epithelium, both in *Gp130-3^d/d^* and *Stat-3^d/d^* mice; however, this was not seen in *Gp130-3^f/f^* and *Stat-3^f/f^* mice [74]. The implantation failure occurred because suppression of uterine gp130 or *Stat-3* is associated with downregulation of some specific P4-responsive genes and a higher uterine estrogenic response under normal ovarian hormone levels. A functional gp130-Stat3 signaling pathway is essential for appropriate uterine E2 and P4 responsiveness for the preparation of the uterus for implantation.

P4 has been administered for multiple clinical purposes, including infertility treatment and contraception. Excessive P4 (4 mg/mouse twice on day 2.5 and 3.5 p.c. [78], 10 mg daily on day 0.5 to 3.5 p.c. [79]) or progestogen (levonorgestrel, 300 μg/kg daily on day 0.5 to 3.5 p.c. [79]) has been administered in mice with inhibited embryo attachment and decidualization through the downregulation of uterine LIF-STAT3 signaling [78,79]. A single dose of excessive P4 (4 mg/mouse twice on day 2.5 p.c.) can suppress *Lif* mRNA expression and phospho-*Stat-3* immunostaining [79].

## 9. Discussion

Approximately 17% of women experience spontaneous pregnancies after successful ART treatment, while around 24% of women have spontaneous pregnancies after failed ART treatment [80,81,82,83,84]. This suggests that at least some women with a history of implantation failure may not have an irreversible defect. The uterine endometrium of fertile women may not be ready for conception in every menstrual cycle, as the maximum efficiency of human natural conception is approximately 30% every menstrual cycle. The uterine receptivity may vary with each menstrual cycle. The HVJ-E vector system may be valuable for understanding the pathophysiology of implantation failure in women because gene suppression using this system does not result into complete knockdown of genes. A limitation of this system is that the efficiency of gene transfection is approximately 75%, even though the same procedure is performed with steady hands.

Local and transient uterine gene transfection of Stat3 decoy or a dominant negative mutant Stat-3 on day 1.5 p.c. suppressed approximately 50% of Stat-3 activity in the uterus on day 5.0 p.c., resulting in <30% implantation with normal serum progesterone levels. Women with unexplained infertility also have normal serum progesterone levels. Implantation failure is a significant cause of infertility in women with unexplained infertility. The Phosphorylated Stat-3 immunostaining is significantly lower in the uterine endometrium during the mid-luteal phase in women with recurrent/repeated implantation failures [85,86] and in women with repeated implantation failure and dormant genital tuberculosis [87] than in fertile control groups, whereas LIF staining intensity did not differ [85]. This suggests that Stat-3 may be an effective diagnostic and therapeutic target for human implantation failure.

The HVJ-E vector system is one of the most efficient gene delivery systems for in vivo murine uteri. Recently, the HVJ-E vector showed intrinsic anticancer activity and has been studied in a clinical trial for the treatment of patients with melanoma [23]. It would be difficult to apply the HVJ-E vector system for infertility treatment because the safety of the HVJ-E vector for the next generation is unclear. However, the HVJ-E vector system using mouse models is a very powerful tool for investigating the pathophysiology of implantation failure in women.

## Figures and Tables

**Figure 1 biomolecules-13-00079-f001:**
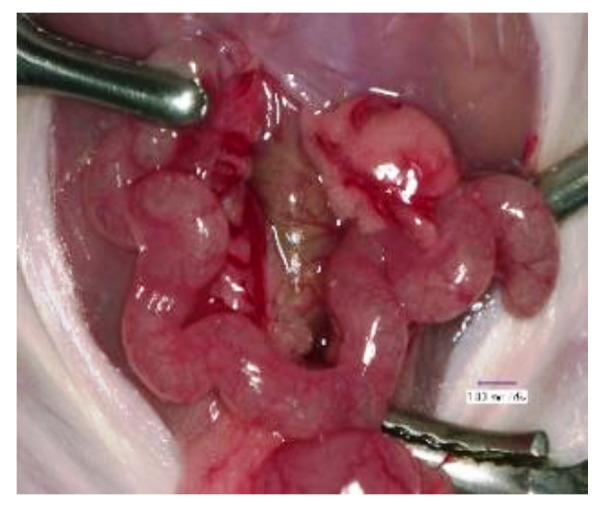
Injection of HVJ-E vector into uterine cavity in mice. The HVJ-E vector suspension (25 μL/horn) was transferred into the uterine cavity using a 30-gauge needle from the oviduct side of the utero-tubal junction on day 1.5 post-coitum. The injection sites and uterine cervix were gently clamped for 10 min. The incision was then closed to allow recovery of the mice.

**Figure 2 biomolecules-13-00079-f002:**
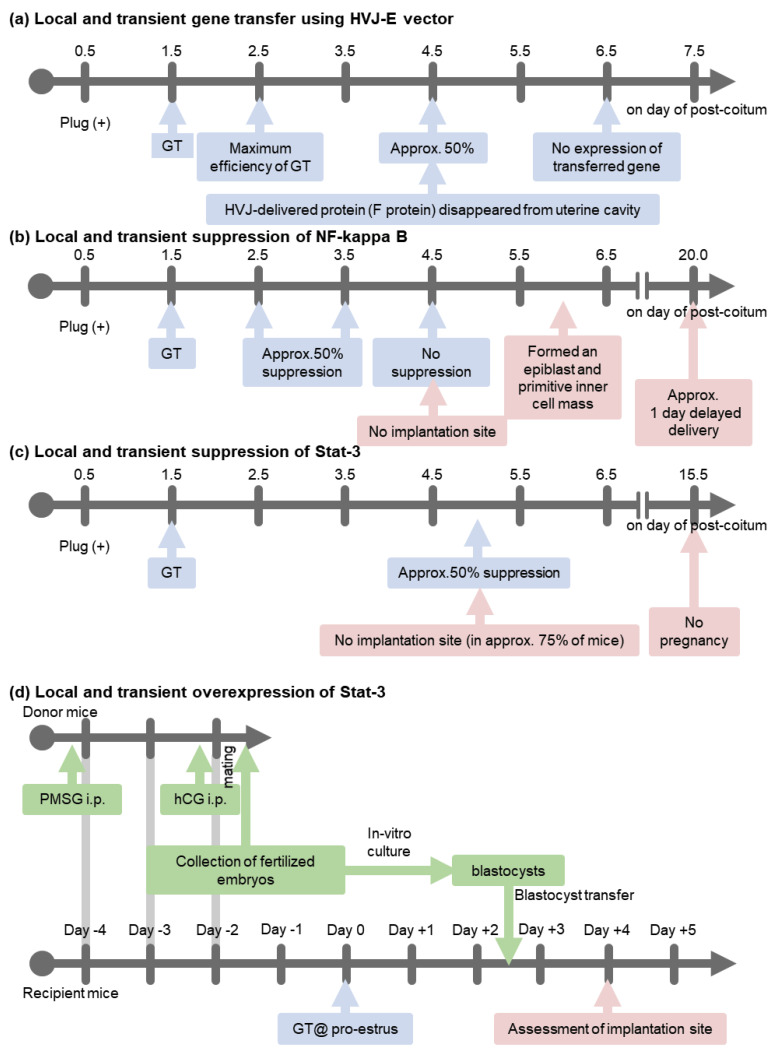
Time course of gene transfection. GT: gene transfer, PMSG: pregnant mare serum gonadotropin, hCG: human chorionic gonadotropin.

**Figure 3 biomolecules-13-00079-f003:**
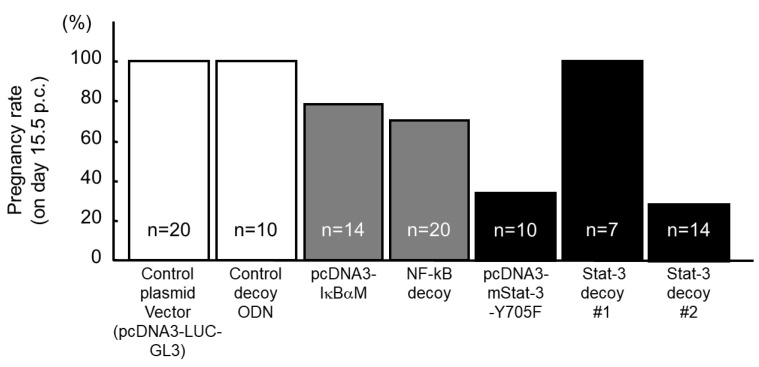
Effect of gene transfer in mouse uterus using HVJ-E vector on pregnancy rate. The dominant negative mutant of NF-κB (pcDNA3-IκBαM), the dominant negative mutant of Stat-3 (pcDNA3-Stat-3-Y705F), or double-stranded decoy ODNs (Table 1) were transferred into the uterine cavity using HVJ-E vector on day 1.5 p.c. On day 15.5 p.c., viable fetuses in utero were assessed in each group.

**Figure 4 biomolecules-13-00079-f004:**
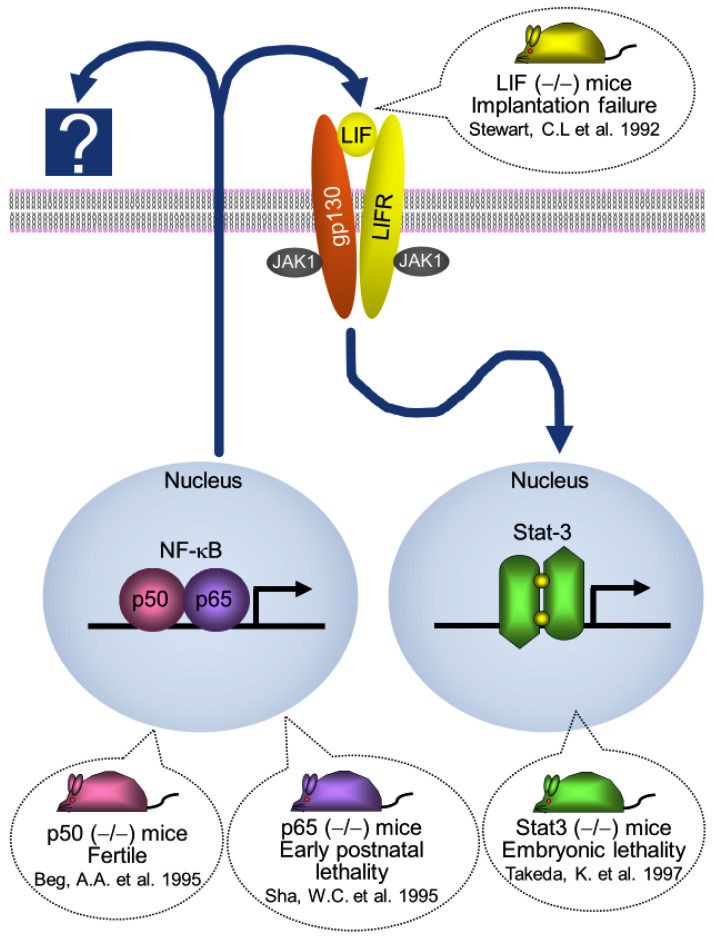
Molecular mechanism of implantation in mouse. NF-κB activation determines the timing of implantation, at least in part, by controlling LIF expression. LIF recruits JAK kinases by binding to LIF receptor (LIFR)-gp 130 and, in turn, recruits the transcription factor Stat-3. NF-κB nuclear factor-kappa B; LIF, leukemia inhibitory factor; JAK, Janus kinase; Stat-3, signal transducer and activator of transcription 3 [38,39,42,43].

**Table 1 biomolecules-13-00079-t001:** Double-stranded decoy ODN sequences.

NF-κB decoy	5’-CCTTGAAGGGATTTCCCTCC-3’ 3’-GGAGGGAAATCCCTTCAAGG-5’
Stat-3 decoy #1	5’-GATCCTTCTGGGAATTCCTAGATC-3’ 3’-CTAGGAAGACCCTTAAGGATCTAG-5’
Stat-3 decoy #2	5’-CCTTCCGGGAATTCCTTCCGGGAATTC-3’ 3’-GGAAGGCCCTTAAGGAAGGCCCTTAAG-5’

Underlines indicate the consensus elements.

## Data Availability

Not applicable.

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
