# Peer review of "An In Vivo Screening Model for Investigation of Pathophysiology of Human Implantation Failure"

_biomolecules, 2022, doi:10.3390/biom13010079_

Round 1

Reviewer 1 Report

Manuscript ID "biomolecules-1884587" entitled " An in vivo screening model for investigation of molecular mechanism of implantation" is an interesting and concise manuscript. 

Despite advances in assisted reproductive technology(ART), "recurrent implantation failure(RIF)" has many challenges. There are various attempts to overcome RIF in clinical practice, but empirical treatments such aspirin, warfarin or immunoglobulin are the mainstay. The implantation of blastocyst into the endometrium in the "receptive phase" appears to be a complex mechanism. 

This review article is based on results in mouse model. It utilizes the HVJ-E vector system to knockout gene of NF-κB and stat-3, and confirms that these gene can affect implantation. Although the results are using mouse models, I hope that further research using this system will be carried out considering that the HVJ-E vector system is being tried for the treatment of tumor such as melanoma. I also expect further research on NF-κB and stat-3.

Author Response

We appreciate your comments. We hope somehow this article can help researchers to get a new idea to improve current ART treatment.

Reviewer 2 Report

Comments in enclosed file

Author Response

We appreciate your comments, which we have read carefully and found very helpful for improving our manuscript. Based on your comments, we have made substantial revisions to our manuscript and want to resubmit it for your consideration. All major changes are highlighted in yellow colour in the revised manuscript.

Reviewer 3 Report

This manucript is a review focused on the method and the results, and has a useful meaning. However, before accept, the writting pattern should be changed little bit. It looks like research pater at early part but later it is review. 

Minor

   In fiugre 3:   put a legend in Y axis

Author Response

(The authors gave the same response as above.)
